# Test-Time Training on Nearest Neighbors for Large Language Models

**Moritz Hardt**
Max Planck Institute for Intelligent Systems, Tübingen
Tübingen AI Center, University of Tübingen

**Yu Sun**
Stanford University

## Abstract

Many recent efforts augment language models with retrieval, by adding retrieved data to the input context. For this approach to succeed, the retrieved data must be added at both training and test time. Moreover, as input length grows linearly with the size of retrieved data, cost in computation and memory grows quadratically for modern Transformers. To avoid these complications, we simply fine-tune the model on retrieved data at test time, using its standard training setup. We build a large-scale distributed index based on text embeddings of the Pile dataset. For each test input, our system retrieves its neighbors and fine-tunes the model on their text. Surprisingly, retrieving and training on as few as 20 neighbors, each for only one gradient iteration, drastically improves performance across more than 20 language modeling tasks in the Pile. For example, test-time training with nearest neighbors significantly narrows the performance gap between a small GPT-2 and a GPT-Neo model more than 10 times larger. Sufficient index quality and size, however, are necessary. Our work establishes a first baseline of test-time training for language modeling.[1]

## 1 Introduction

Machine learning traditionally separates training and testing. Once trained, a model remains frozen during evaluation. But nearly as old as machine learning is the intriguing idea to update the model at test time with data relevant to each individual test instance. Variants of this idea have existed for almost fifty years, including locally weighted regression (Stone, 1977; Cleveland, 1979), local learning (Bottou & Vapnik, 1992), and SVM-KNN (Zhang et al., 2006). Recently, test-time training has grown again in popularity with the rise of deep learning, exposing a large space of possible heuristics (Krause et al., 2018; 2019; Sun et al., 2020).

We investigate a simple, yet powerful, heuristic in this space, called *test-time training on nearest neighbors* (TTT-NN), for the task of language modeling. For each test instance, we retrieve its nearest neighbors from a huge database, and fine-tune the model on those neighbors before applying it to the test instance.

Fine-tuning language models is a well known practice to boost its performance on specific domains or tasks. However, there seems to be no consensus on what exactly defines a domain or task (Gururangan et al., 2021), and conventional boundaries between previously defined domain or tasks have become increasingly unclear with the recent advances in general-purpose language models (Brown et al., 2020; Bubeck et al., 2023). From the perspective of test-time training, each test instance defines its own "domain". Our hope is that a sufficiently large database will contain enough data relevant to each "domain" induced by a test instance, such that fine-tuning improves performance locally.

### 1.1 Our Contributions

We present a system for test-time training on nearest neighbors, based on a large-scale index as illustrated in Figure 1. Our distributed index can serve each nearest neighbor query to approximately 200 million vectors and 1TB of data in approximately one second on standard hardware. The vectors represent text embeddings of all training sequences in the Pile dataset (Gao et al., 2020).

---

[1] Code, index files, and model checkpoint: `https://github.com/socialfoundations/tttlm`.

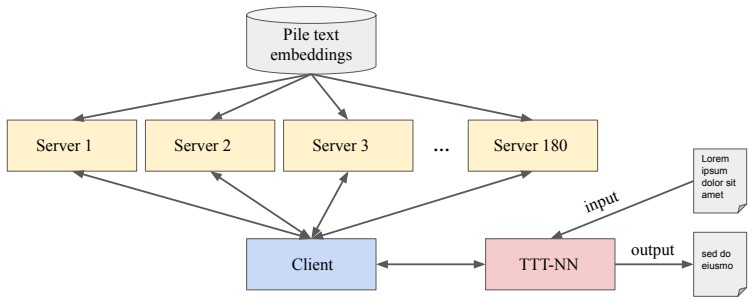

Figure 1: System architecture for test-time training with nearest neighbors (TTT-NN).

We evaluate our method on all 22 tasks for language modeling from the Pile benchmark. We focus on three causal language models of increasing size: a small GPT-2 model with 117M parameters, a large GPT-2 model with 774M parameters, and a GPT-Neo model with 1.3B parameters. We find that training for only one gradient iteration on as few as 50 neighbors reduces a normalized perplexity measurement, the *bits per byte* metric, by 20%. In fact, most of the gain is already realized after only 20 neighbors. When it comes to some tasks, especially code generation as in the `pile_github` task, the bits per byte metric goes down by more than 60% during test-time training. See Figure 2.

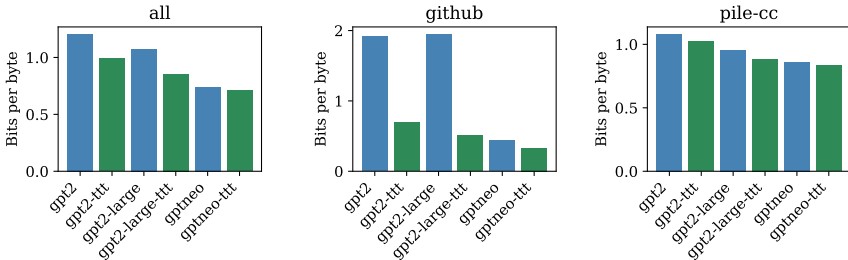

Figure 2: Comparison of different models before and after TTT-NN with 50 neighbors. Left: All Pile tasks. Center: Best performing Pile task (`pile_github`). Right: Largest Pile task (`pile_pile-cc`).

The GPT-Neo model was specifically trained to convergence on the entire Pile dataset. In contrast, GPT-2 was trained only on some subsets of the dataset, roughly `pile_pile-cc`, but not on others such as `pile_github`. Therefore, we can see how TTT-NN on a model that was not pre-trained on a specific Pile task stacks up to a model that was. We see that the improvements due to test-time training on tasks unseen during training, such as `pile_github`, can be dramatic. TTT-NN still helps on seen tasks, such as `pile-cc`, but the improvements are more moderate.

Our results suggest that test-time training can increase the effective capacity of a model. This comes at the obvious cost of increased inference time, like other retrieval augmented techniques. While this cost may be prohibitive for some real-time applications, test-time training can be stopped at any time and still yield a significant improvement (Figure 6). It is also less of a concern in other applications, such as offline code generation and synthesis of sequences for scientific purposes. From this perspective, it is encouraging, and worth investigating further, that test-time training performs best on code generation within the Pile.

## 2 RELATED WORK

### 2.1 LOCAL LEARNING

As discussed at the beginning, the idea of using local data to train a parametric model has a long history and many variants, depending on how locality is defined and what the model is. The oldest literature in nonparametric linear regression (Stone, 1977) and locally weighted regression (Cleveland, 1979; Cleveland & Devlin, 1988; Ruppert et al., 1995; Atkeson et al., 1997) uses a linear model and engineered distance

metrics, e.g. kernels, to assign more weights to training data closer to each test instance. Because initialization does not affect the optimal solution of linear regression, each test instance trains a model from scratch. SVM-KNN (Zhang et al., 2006) applies the same idea to support vector machines.

The most relevant paper is Bottou & Vapnik (1992). Their method, called local learning, first trains a convolutional neural network on images of handwritten digits. The local training data here are chosen to be the nearest neighbors to the test instance, in terms of Euclidean distance in pixel space. Because it is slow to train the entire network for many epochs on each subset of neighbor, they instead only train the last linear layer of the network, as a trade-off between computational cost and performance gain. In comparison, our method can afford to fine-tune the entire network, using its standard training setup, because each neighbor only needs one gradient iteration.

## 2.2 Language Models using Retrieval

Many recent advances in language models retrieve nearest neighbors from the training set, but use them differently from our approach. The most successful line of work uses those neighbors as additional context, also known as feature augmentation, for the test instance. Some prominent examples are Guu et al. (2020), Lewis et al. (2020), Borgeaud et al. (2022), Wang et al. (2022b), Wang et al. (2022b), and Izacard et al. (2022). The success of these methods at test time requires having the retrieved neighbors as additional context also during training, mirroring the procedure at test time. In comparison, our method requires no special training; we simply downloaded pre-trained models from HuggingFace. In Subsection 5.1, we evaluate against a baseline that uses neighbors as context at test time without special training.

At test time, our method requires one forward and backward pass through each of the neighbors individually. Using the neighbors as additional context requires only one forward pass through the neighbors, but the time complexity of self-attention is quadratic in the context length, consequently the number of neighbors. With 50 neighbors, the quadratic cost can be significant. Our method does not have this concern.

Another related line of work is KNN-LM (Khandelwal et al., 2019; He et al., 2021; Alon et al., 2022). The basic idea is that autoregressive language modeling maps each context window of tokens to a predicted distribution over the next token. This can alternatively be achieved with a nonparametric approach: find all tokens in the training data whose context window is similar to the test instance, and use their (weighted) histogram as the predicted distribution. Khandelwal et al. (2019) simply uses a convex combination of the predicted distributions from the two approaches as the final prediction. In Subsection 5.1, we evaluate against a variant of KNN-LM that is computationally feasible for the Pile.

## 2.3 Test-Time Training and Dynamic Evaluation

Beside the ones mentioned in Subsection 2.1, many papers explore the broader idea of training a different model specific to each unlabeled test instance. An early attempt is transductive learning, beginning in the 1990s (Gammerman et al., 1998). Vapnik (2013) states the principle of transduction as: "Try to get the answer that you really need but not a more general one." This principle has been most commonly applied on SVMs (Vapnik, 2013; Collobert et al., 2006; Joachims, 2002) by using the test data to specify additional constraints on the margin of the decision boundary.

More recently, the idea of test-time training has been applied to deep learning, for generalization under distribution shifts (Sun et al., 2020; Gandelsman et al., 2022; Wang et al., 2022a; Sun et al., 2023). It produces a different model for every single test input through self-supervised learning. Another line of work called dynamic evaluation (Krause et al., 2018; 2019) fine-tunes the language model on the context of the test instance. In Subsection 5.1, we compare with a dynamic evaluation baseline.

## 3 Nearest Neighbor Index

Our nearest neighbor index is based on text embeddings of the Pile training set. The training set is split across 30 files, each approximately 7M sequences and 43GB of text. The entire dataset has 210M sequences and size 1.3TB. In addition, the Pile dataset has a validation set and a test set that we do *not* include in the index.

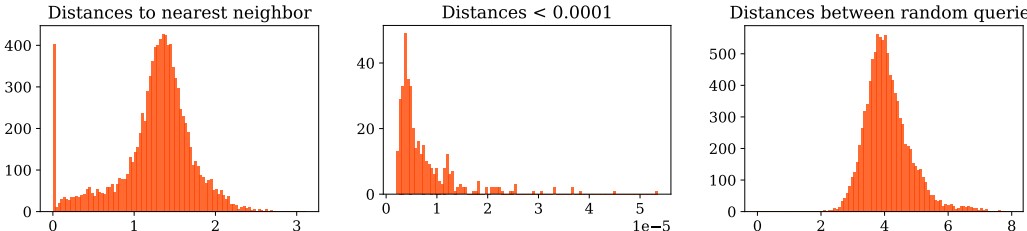

Figure 3: Left: Histogram of distances to nearest neighbor for $10,000$ random queries from the validation set. Center: Focusing on the $400$ smallest distances. Right: Distances between $10,000$ pairs of random queries.

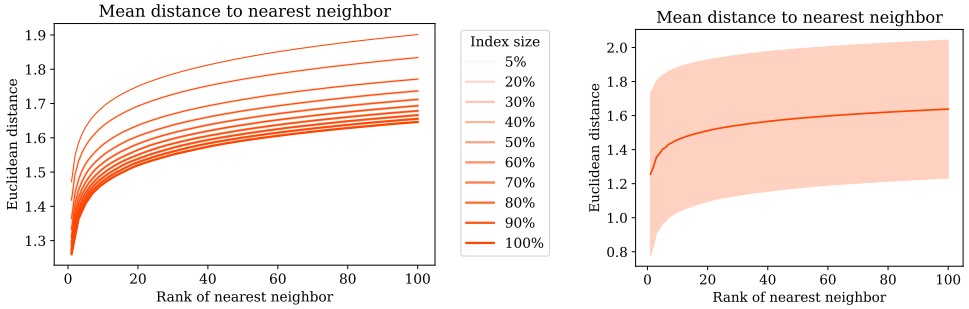

Figure 4: Left: Mean distances to the nearest neighbor as index size grows. Right: Mean distances with standard deviation shown as shaded region for the full size index.

The text embedding model we use is based on a large Roberta model (Liu et al., 2019) with 355M parameters. Starting from a pre-trained Roberta model, we train the model on the Pile for 1.7M iterations, with the standard batch size of 128, corresponding to approximately one pass over the training set. The embedding dimension is 1024. Training the embedding model on the entire training set proved to be important for index quality. To build the index, we apply the embedding model to each sequence in the training set. For simplicity, we naively truncate long sequences to the maximum sequence length of the embedding model. In other words, on long sequences we select neighbors based on only a prefix of the sequence.

We store the text embeddings in a nearest neighbor data structure. As a proof of concept, we simply use the FAISS Flat L2 index (Johnson et al., 2019). The entire index is 810GB, adding up to 2.1TB for data and vectors combined. To speed up query time across the data structure, we build a distributed client-server architecture, illustrated in Figure 1, on top of the FAISS index. The system splits the index across 180 servers each running on one chunk of the index and dataset. A client sends out nearest neighbor queries to each server and receives the respective neighbors in each chunk. The client then builds a local nearest neighbor structure out of the received results and queries the local structure for the final results.

The distributed index can answer a query to the entire Pile dataset in around one second, including time to compute the query embedding, on standard hardware and network access without any GPU acceleration on the server side, which would further accelerate query time. Specifically, we observed a mean query time of 1.35 seconds with a standard deviation of 0.12 seconds, computed across 1000 random queries form the Pile training set. The distributed architecture gives a flexible way to split the index across different hardware footprints. Increasing the number of servers decreases the local query processing time on each server, but increases the time for network traffic. We find that 180 servers is a reasonable trade-off for the cluster we use.

Figure 3 takes a closer look at nearest neighbor distances. We pick $10,000$ random points from the validation set and retrieve their neighbors from the index. The index does not include the validation or test set. However, there is a small cluster of about $4\%$ of the points that have a near exact match in the index. In general, distances to the neighbors are substantially smaller than distances between the embeddings of two random queries, as the right panel of the figure shows. Figure 4 shows that distances to nearest neighbors continue to decline as we increase the index size. This reflects the benefits of using a larger index.

## 4 TEST-TIME TRAINING ON NEAREST NEIGHBORS

Although the idea of test-time training on nearest neighbors is simple and intuitive, making it work involves a few important design choice.

The first is how exactly to train on the neighbors. Our best practice trains on each neighbor sequentially in *increasing* distance, starting from the very nearest neighbor. The opposite order of going from farthest to nearest performs significantly worse. This might seem counter-intuitive. A priori, it could have been plausible that we should train on the nearest last, because it contains the most relevant information that the model should not forget. But the opposite turns out to be the case. One interpretation of this empirical phenomenon is that, it could be advantageous to take the largest gradient step at the beginning on the best available data point, and therefore ground the fine-tuning process in a better region of the loss landscape. We also experiment with a much more costly variant: put all neighbors into a batch, and take gradient steps on this entire batch of data repeatedly. This is much slower than the sequential method, but does not lead to significant improvements.

The second design choice considers the many sequences in the Pile dataset that exceed the maximum sequence length of the base models. In such cases, we split the long sequences into chunks, of length equal to the model's maximum sequence length. This means that a single retrieved neighbor can result in several gradient updates. We again iterate over these chunks sequentially, one gradient update per chunk.

After fine-tuning the model on the current test instance, we reset its trainable parameters to their original states - those of the pre-trained base model. We could potentially let changes to the model accumulate during the entire evaluation run, but intuitively this would only make sense under some additional assumptions on temporal smoothness, that the next test instance is similar to the current one in some sense.

**Hyper-parameters – or lack thereof.** Beyond these design choices, the method requires no hyper-parameter tuning. A remarkable aspect is that we can simply reuse the default hyper-parameters for the model and the optimizer available for each model in the HuggingFace library. This is in contrast with other test-time augmentation methods that often invest labor and computational resources in tuning the augmentations.

## 5 RESULTS

We focus on language modeling tasks with causal language models, specifically, models in the GPT family. When it comes to language modeling, the most widely studied metric has been perplexity. But there are actually several different ways researchers have computed and reported perplexity. This is, in part, due to the computational burden of evaluating perplexities exactly across large corpora. To ease reproducibility of our numbers, we use Eleuther-AI's `lm-evaluation-harness` library (Gao et al., 2021) for evaluating perplexities on the Pile. We report the *bits per byte* metric as recommended by Gao et al. (2020). This metric is proportional to the negative log-likelihood loss normalized by a dataset-specific constant. The Pile test set has $214, 584$ sequences. For efficiency, we evaluate on $20\%$ of the test set, corresponding to $42, 916$ sequences. We defer bootstrap error bars to the supplementary materials as they are generally small and do not affect the interpretation of the results.

We first evaluate on a small GPT-2 model with 117M parameters - the default HuggingFace `gpt2` model from the `transformers` library. See Figure 5. Figure 6 showcases the decrease in three different perplexity measures as we increase the number of nearest neighbors to train on. We can see that using 20 neighbors already achieves most of the decrease in perplexity, computationally costing less than half than using all the neighbors. Using additional neighbors continues to decrease perplexity. Figure 6 focuses on the six largest Pile tasks in increasing order. These six tasks combined make up more than $70\%$ of the Pile benchmark.

Figure 7 summarizes our results for a large GPT-2 model with 774M parameters, specifically, the Huggingface `gpt-large` model. Finally, we evaluated our method on the GPT-Neo model with $1.3B$ parameters by Eleuther-AI, that aims to replicate a small GPT3 model. Note this model is specifically trained on the Pile, so it gives us an important insight into how test-time training compares with pre-training on the entire dataset. What we can see from Figure 2 and Figure 8 is that remarkably, TTT-NN on the GPT-2-large model can recover most of the gains of pre-training a model twice as large to convergence on the full dataset.

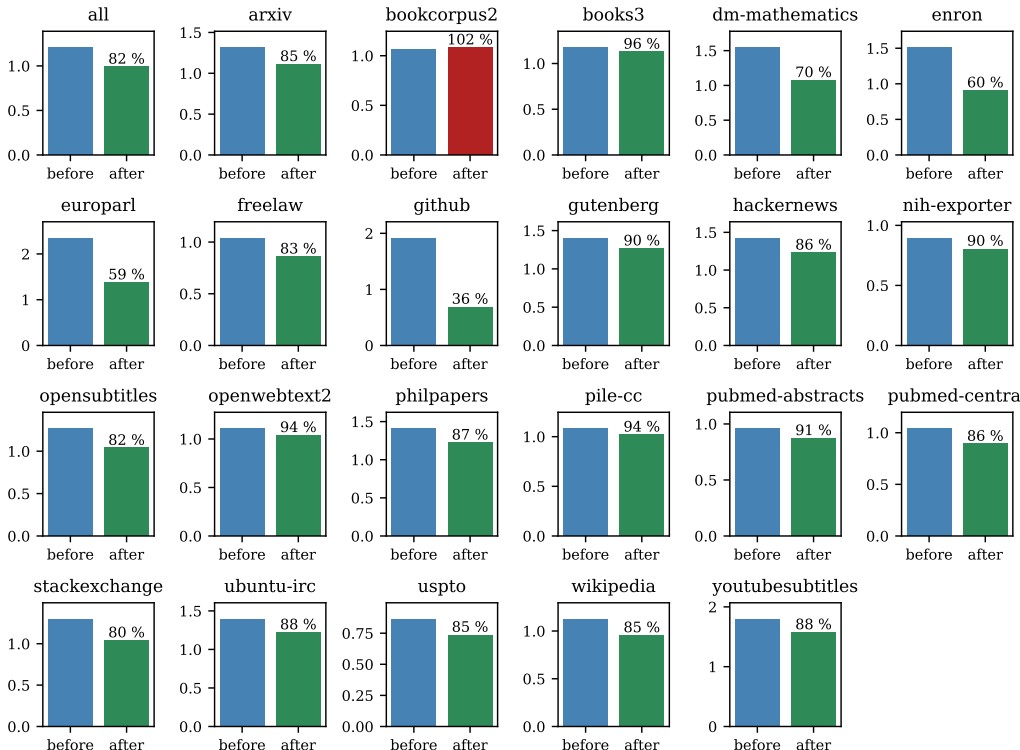

Figure 5: Bits per byte results on all Pile tasks for a small GPT-2 model (117M parameters) before and after test-time training on 50 nearest neighbors.

**Splitting sequences to avoid retrieval-evaluation overlap.** There is an important subtlety. In our standard protocol, we use the same sequence for retrieving nearest neighbors as we evaluate on. This raises the concern that we are training on sequences that were chosen with knowledge of the target of prediction. We address this concern by splitting the evaluation sequence into two chunks of equal length, using the first for retrieval and the second for evaluation. It turns out that doing so changes our results very little, for each task by typically less than 1% and rarely up to 2%. On the entire `pile_all` task, the bits per byte number is 83%, up from 82% for the variant without splits. The supplementary materials contain the detailed breakdown of numbers for the splitting protocol. Since the split is somewhat arbitrary, we present the results with no split in the main body of the paper.

**Training costs.** Training costs vary greatly depending on the length of the retrieved sequences. Some tasks have book-length sequences, while others consist of very short sequences. Recall that long sequences are split into multiple chunks during fine-tuning. This leads to a highly heterogeneous training cost profile across different tasks as illustrated in Figure 9. Even within tasks there is significant variation. The figure shows training cost in seconds per neighbor on a single NVIDIA A100 GPU.

Needless to say, test-time training significantly increases the compute footprint at inference time. For some applications, running all 50 iterations may be prohibitive. However, as shown in Figure 6, those gradient iterations can be stopped at any time and still yield a significant improvement, with the first few steps being the most helpful, thanks to the sequential order discussed at the beginning of this section. In addition, there are many applications of language modeling that are not real-time, such as generating sequences for scientific applications, or offline code synthesis.

## 5.1 Comparisons with Baselines

We compare TTT-NN with three natural baselines: 1) in-context neighbors, 2) interpolation with the distribution of tokens among the neighbors, and 3) dynamic evaluation. We found that our method outperforms all three.

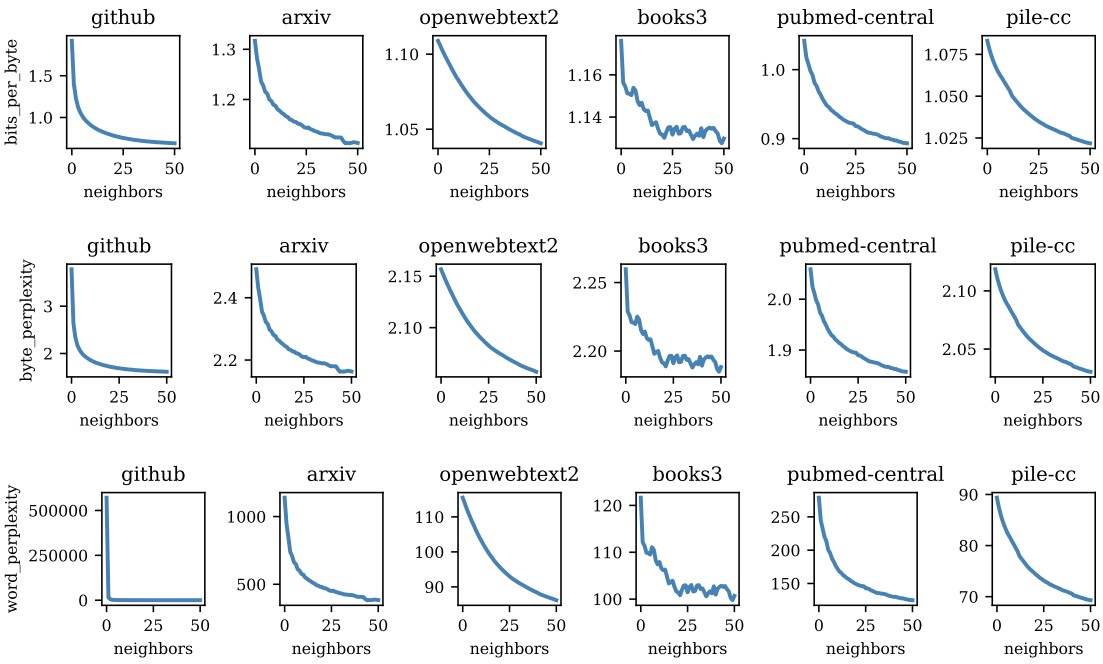

Figure 6: How different measurements of perplexity, as provided by Gao et al. (2020), decrease with the number of neighbors on the six largest Pile tasks in ascending order, for GPT-2-Small. Top: Bits per byte. Center: Byte perplexity. Bottom: Word perplexity.

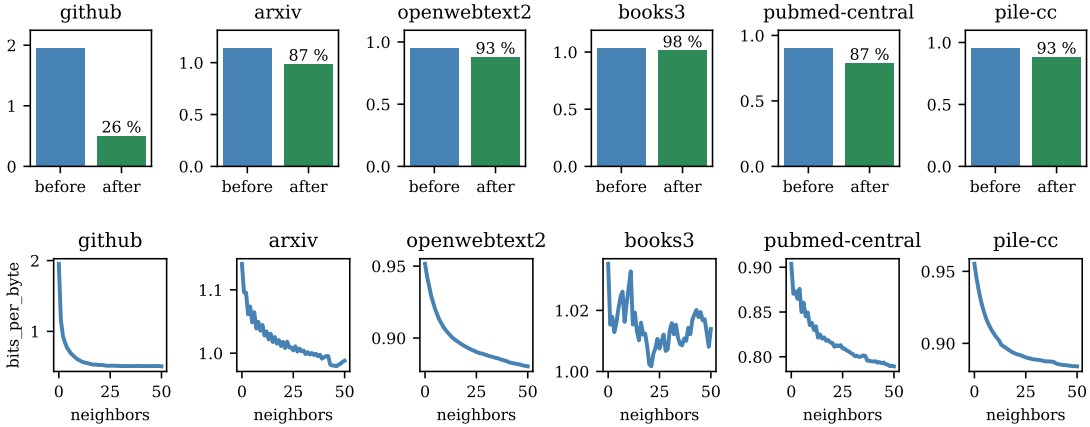

Figure 7: Results for GPT-2-Large. Top: Before and after TTT-NN with 50 neighbors on the top six tasks. Bottom: How the bits per byte metric decreases with additional neighbors.

Table 1 shows detailed results on the large GPT-2 model. Section 2 discusses those baselines in the context of related work. The next few paragraphs discuss the implementation of those baselines.

Many methods use retrieved neighbors as additional context for the test instance, without test-time training. Unlike ours, models for those methods need to be trained also with retrieval. Due to the prohibitive cost of training with retrieval, we experiment with a baseline that simply uses the neighbors in-context at test time. Specifically, we concatenate the neighbors in increasing distance, and include as much of the concatenated text as possible into the context window of the test instance, in addition to its original context. This baseline improves performance in a few cases, e.g. `pile_enron`, but does not help much overall.

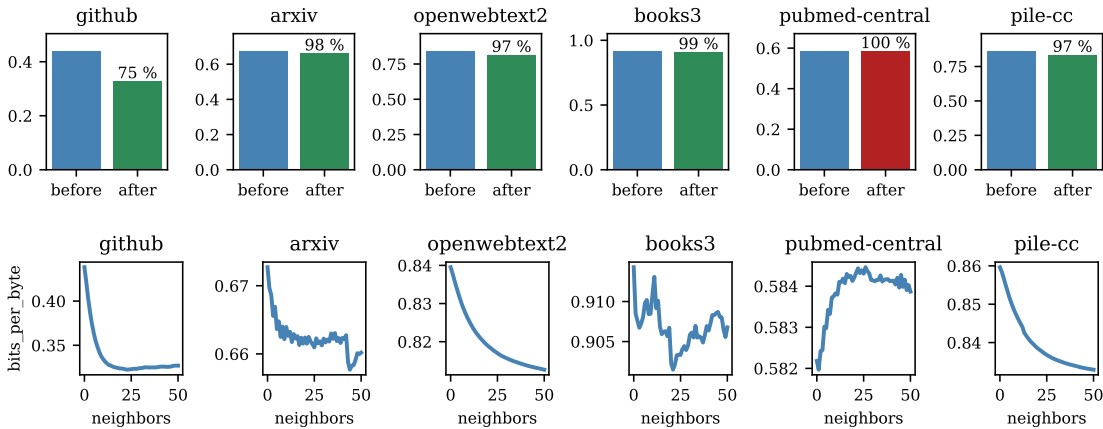

Figure 8: Results for GPT-Neo (1.3B parameters). Top: Before and after TTT-NN with 50 neighbors on the top six tasks. Bottom: How the bits per byte metric decreases with additional neighbors.

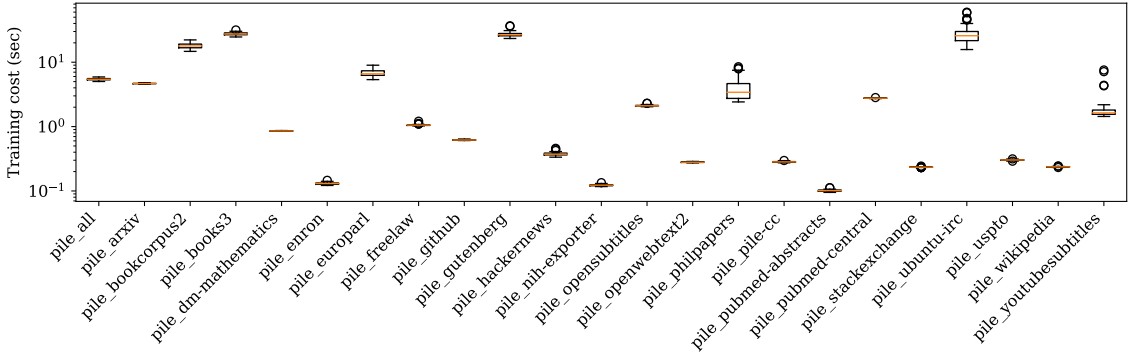

Figure 9: Training costs in seconds per neighbor on each task.

Also discussed in Subsection 2.2, the KNN-LM line of work interpolates between the distribution of the next token among the neighbors, and the distribution predicted by the language model. However, the cost of directly implementing their method on large datasets like the Pile is prohibitive. While our method creates a database entry for each document, theirs creates one for each token in the training data; also, retrieval needs to be performed for each token in the test instance.

To make the comparison feasible, we propose a modification. For each next token in the test instance, instead of using a different distribution by querying with the previous tokens as context, we can make use of our document-level database by querying with all tokens in the test instance. The rest remains the same: We interpolate this distribution with the one predicted by the language model, and choose the best interpolation hyper-parameter on `pile_all` (2% NN, 98% LM). In Table 1, this modified baseline is called *Interpolate*.

Finally, we experiment with a dynamic evaluation baseline, discussed in Subsection 2.3. Because dynamic evaluation directly trains on the text that would otherwise be the target of prediction, this baseline only makes sense in settings where the text for test-time training and evaluation do not overlap. The last paragraph of Section 5 describes such a setting for our method to avoid retrieval-evaluation overlap: the sequence of tokens in the test instance are split in two halves, the first half for retrieval, and the second for evaluation.

Applying dynamic evaluation to this setting, we instead use the first half for test-time training, allowing for as many as 50 iterations. However, dynamic evaluation starts overfitting after 7 iterations. We therefore report the results with 7 iterations. We observe that this baseline leads to meaningful improvements in many tasks,

Table 1: Comparison with baselines. All results are on GPT-2-Large. Results for *TTT-NN (Ours)* correspond to those labeled *after* in Figure 7, and for *Base Only* correspond to *before*. The baselines are detailed in Subsection 5.1. *In-Context* simply adds the retrieved neighbors to the context. *Interpolate* is a modified version of KNN-LM (Khandelwal et al., 2019). *Dyn. Eval.* is dynamic evaluation (Krause et al., 2018).

| Task | In-Context | Interpolate | Dyn. Eval. | TTT-NN | Base Only |
|---|---|---|---|---|---|
| pile_all | 1.06 | 0.99 | 1.03 | **0.85** | 1.07 |
| pile_arxiv | 1.14 | 1.13 | 1.13 | **0.99** | 1.14 |
| pile_bookcorpus2 | 0.95 | 0.96 | **0.91** | 1.00 | 0.96 |
| pile_books3 | 1.03 | 1.03 | 1.09 | **1.01** | 1.04 |
| pile_dm-mathematics | 1.42 | 1.42 | 1.30 | **0.99** | 1.43 |
| pile_enron | 1.28 | 1.16 | 1.38 | **0.72** | 1.44 |
| pile_europarl | 1.96 | 1.92 | 2.13 | **1.15** | 1.96 |
| pile_freelaw | 0.88 | 0.85 | 0.85 | **0.72** | 0.89 |
| pile_github | 1.88 | 1.06 | 1.29 | **0.51** | 1.95 |
| pile_gutenberg | 1.20 | 1.17 | **1.14** | 1.18 | 1.21 |
| pile_hackernews | 1.28 | 1.26 | 1.16 | **1.09** | 1.30 |
| pile_nih-exporter | 0.75 | 0.78 | 0.86 | **0.63** | 0.78 |
| pile_opensubtitles | 1.15 | 1.15 | 1.18 | **0.75** | 1.15 |
| pile_openwebtext2 | 0.95 | 0.95 | 1.00 | **0.88** | 0.95 |
| pile_philpapers | 1.20 | 1.36 | 1.38 | **1.07** | 1.22 |
| pile_pile-cc | 0.94 | 0.95 | 0.99 | **0.88** | 0.96 |
| pile_pubmed-abstracts | 0.77 | 0.81 | 0.92 | **0.74** | 0.82 |
| pile_pubmed-central | 0.90 | 0.90 | 0.89 | **0.79** | 0.90 |
| pile_stackexchange | 1.07 | 1.03 | 1.05 | **0.88** | 1.09 |
| pile_ubuntu-irc | 1.27 | 1.27 | 1.30 | **1.13** | 1.27 |
| pile_uspto | 0.73 | 0.75 | 0.78 | **0.57** | 0.75 |
| pile_wikipedia | 0.92 | 0.95 | 1.03 | **0.81** | 0.98 |
| pile_youtubesubtitles | 1.55 | 1.41 | **1.34** | 1.42 | 1.51 |

and in three cases better than ours. It is especially helpful, although not as much as ours, for pile_github. Note that the same heuristic can also be added to our method: Our first iteration can simply be taken on the query text – the test-time training data for dynamic evaluation, with the remaining iterations on the neighbors.

## 6 LIMITATIONS AND FUTURE WORK

Our work shows that test-time training with nearest neighbors can boost the effective capacity of a model in terms of perplexity for language modeling tasks. But it does not show if test-time training also exhibits some of the emergent phenomena associated with larger models. Broader evaluation on other non-perplexity tasks is an important direction for future work. Code generation seems to be a particularly promising direction due to the strong results of TTT-NN on pile_github. It would be interesting to conduct studies of end-to-end code synthesis tasks in this direction. Furthermore, we would like to understand if this strength extends to life sciences applications, such as those involving large-scale medical or protein databases.

Also worth investigating is the use of test-time training in the context of fairness and safety. Our approach might help mitigate biases against under-represented groups in the data source that large language models are trained on. As long as the database contains some high-quality data from a particular under-represented group, TTT-NN could potentially benefit this group at least as much as the larger groups. Our experiments on the Pile give some support for this intuition, as the smaller tasks with less than 1% size, e.g. pile_enron and pile_europarl, see much more significant improvements than the larger ones, e.g. pile_pile-cc and pile_pubmed-central, with more than 30% size together. Test-time training might also help mitigate adversarial behaviors, including data poisoning attacks, by superimposing data at test time from a trusted data source. We hope to see more future work in this direction.

## ACKNOWLEDGEMENTS

We are grateful to Sasha Rush for tremendously helpful conversations at an early stage of the project. We also thank Carsten Eickhoff and Tatsunori Hashimoto for helpful feedback and suggestions. Yu Sun would like to thank his other PhD advisor, Alexei A. Efros, his internship manager at Meta, Xinlei Chen, and his friend, Beidi Chen, for their help in the general idea of this project. Yu Sun is supported in part by Oracle Cloud credits and related resources provided by the Oracle for Research program.

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

## A  RESULTS FOR GPT-2-SMALL ON SPLIT SEQUENCES

Here we show the results where we split each sequence in half, using the first part for retrieval and the second part for evaluation. If the sequence is more than twice the maximum sequence length, then our first part has length equal to the maximum sequence length, and the second part is the rest. For these results, we focus on the small GPT-2 model.

The model is available on HuggingFace at https://huggingface.co/gpt2. We use a learning rate of 2e-5 for the Adam optimizer with $\epsilon$ value 1e-08. The maximum sequence length of the model is 1048 tokens.

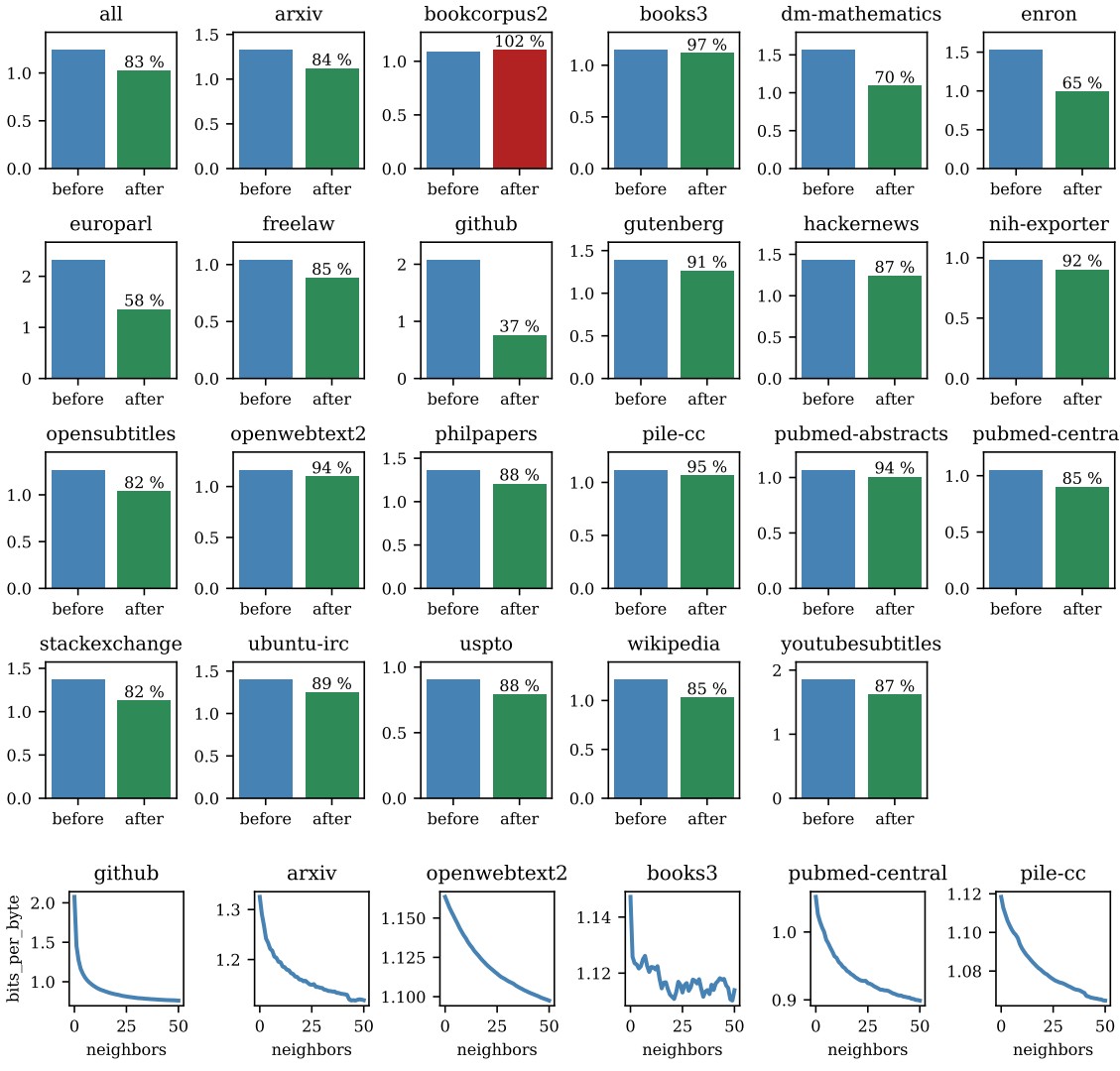

Figure 10: Bits per byte results on all Pile tasks for GPT-2-Small (117M parameters) before and after training on 50 nearest neighbors, where we split the sequence into two non-overlapping halves for query and evaluation.

## B  ALL RESULTS FOR GPT-2-LARGE

The model is available on HuggingFace at `https://huggingface.co/gpt2-large`. We use a learning rate of `2e-5` for the Adam optimizer with $\epsilon$ value `1e-08`. The maximum sequence length of the model is 1048 tokens.

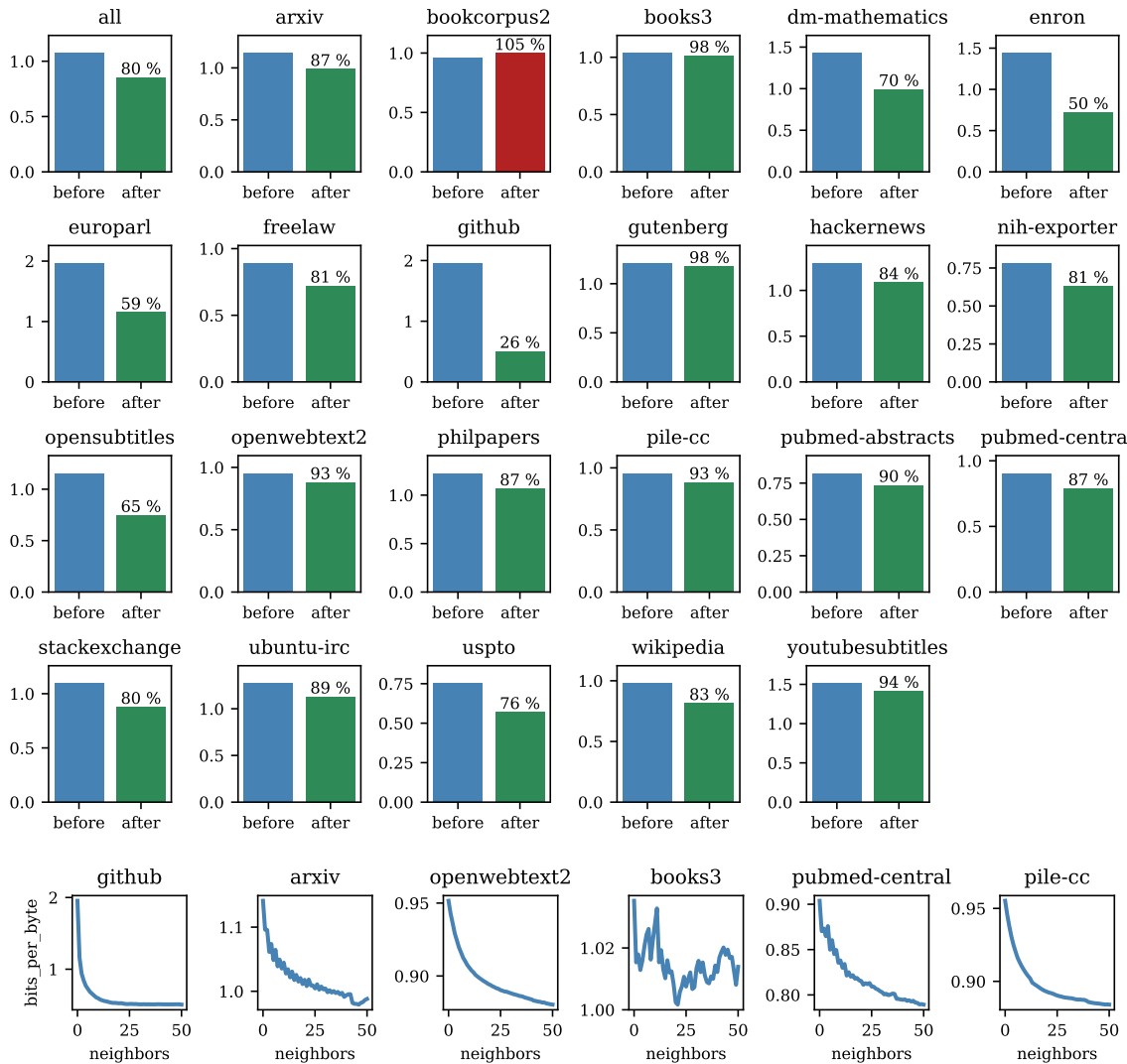

Figure 11: Bits per byte results on all Pile tasks for GPT-2-Large (771M parameters) before and after training on 50 nearest neighbors.

## C ALL RESULTS FOR GPT-NEO

The model is available on HuggingFace at https://huggingface.co/EleutherAI/gpt-neo-1.3B. We used a learning rate of 5e-6 for the Adam optimizer with $\epsilon$ value 1e-08. The maximum sequence length of the model is 2048 tokens.

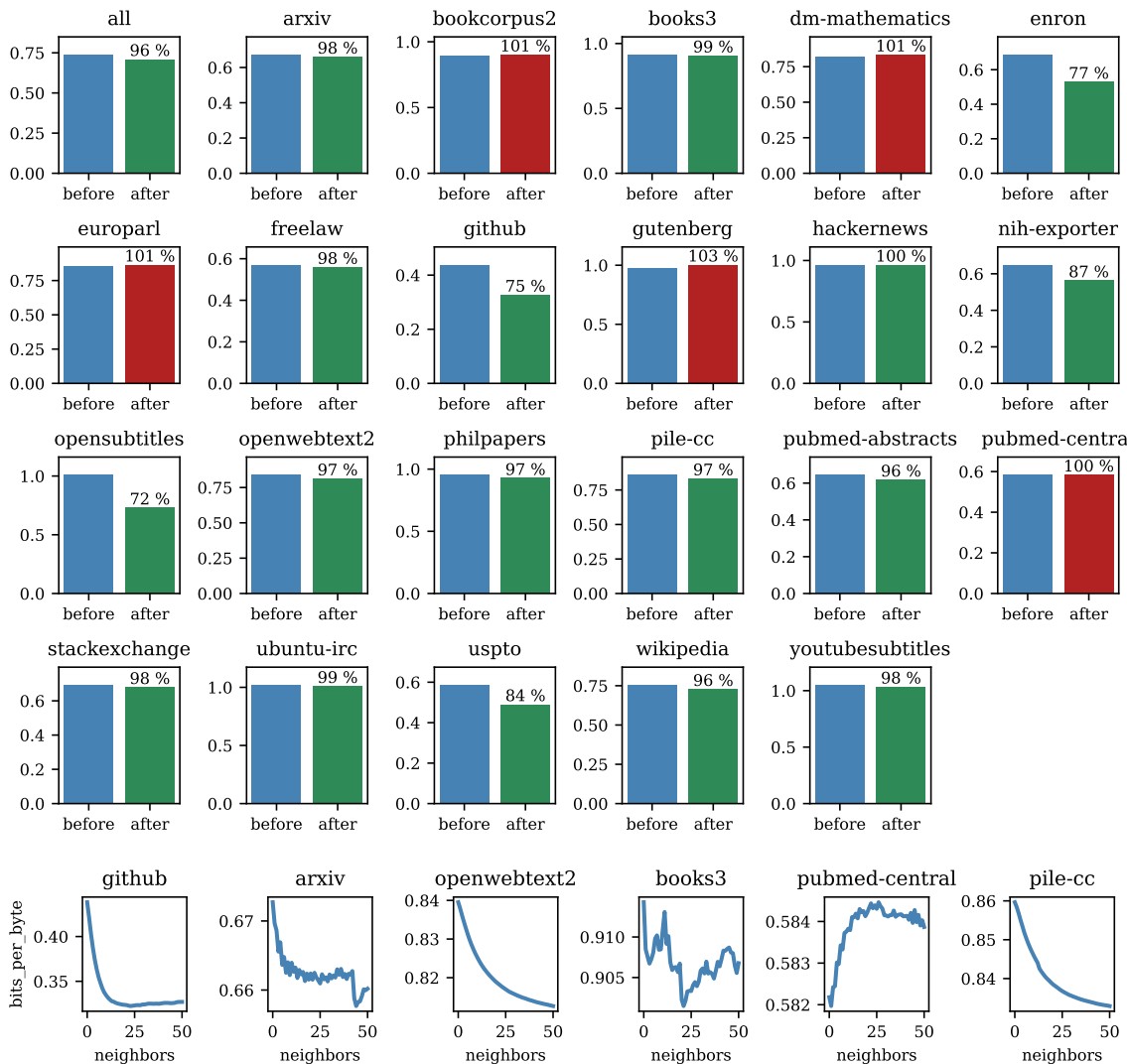

Figure 12: Bits per byte results on all Pile tasks for GPT Neo (1.3B parameters) before and after training on 50 nearest neighbors.

# D    BOOTSTRAP ERROR BARS

In this section, we report empirical bootstrap error bars for our bits per byte evaluation metrics. Computing valid error bars for perplexity metrics is non-trivial for two reasons. First, perplexities can assume large values and are generally unbounded to the top. Second, the metric is an exponential of a sum of values, rather than a sum of values for individual data points as would be the case with a metric like accuracy. For these reasons, we follow established practices in statistics and report empirical bootstrap error bars.

For a given task consisting of $n$ sequences, a bootstrap sample will sample the individual negative log likelihoods on each sequence i.i.d. with replacement $n$ times. For each bootstrap sample we compute the bits per byte metric. Error bars then refer to quantiles of the bits per byte values. We choose 1000 bootstrap samples and report the $(0.1, 0.9)$ quantiles as lower and upper error bars, respectively.

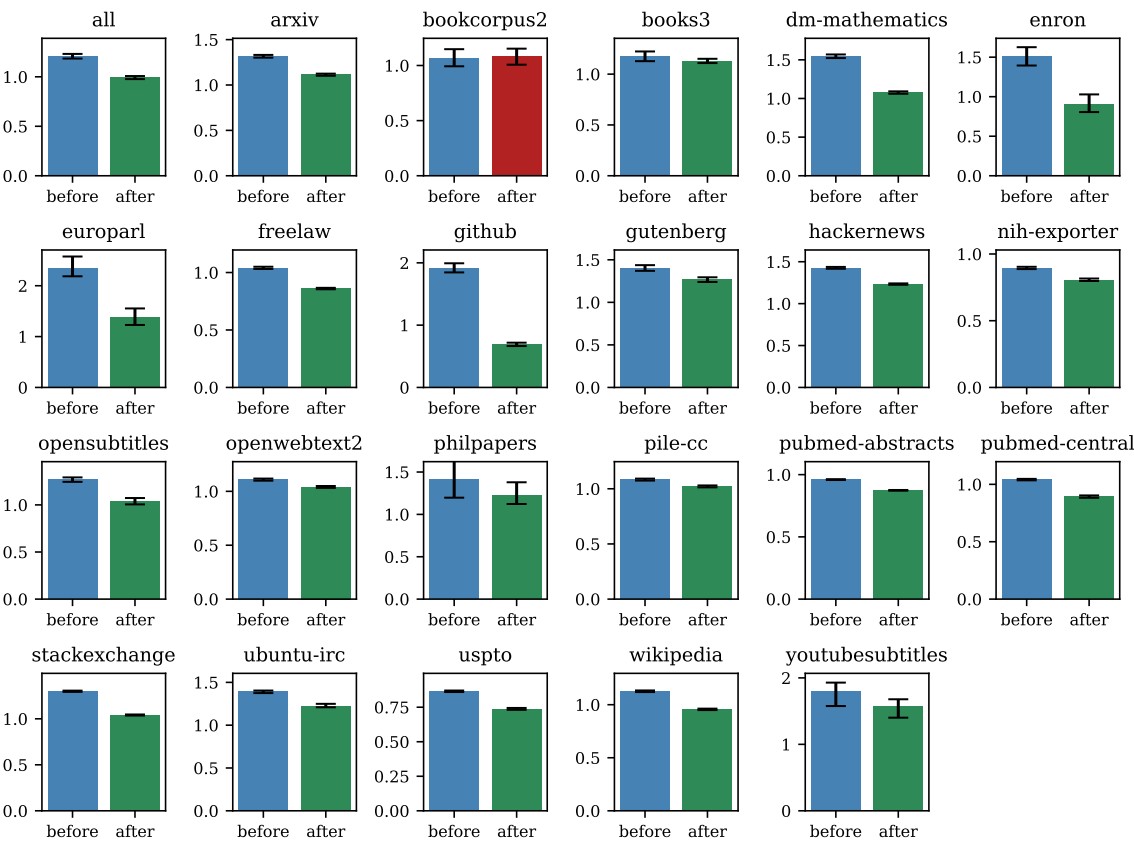

Figure 13: Bits per byte results on all Pile tasks for GPT-2-Small (117M parameters) with error bars indicating 0.1 and 0.9 quantiles across 1000 bootstrap samples. This means only $10\%$ of bootstrap samples fall outside the error bars to the top and bottom, respectively.

The error bars for the larger models are only smaller, since the perplexities are generally smaller and have fewer large values. We therefore omit the corresponding plots for GPT-2-Large and GPT-Neo.

# E  TRAINING COSTS

For completeness, we compare all training costs below across all tasks and the three models we evaluated on.

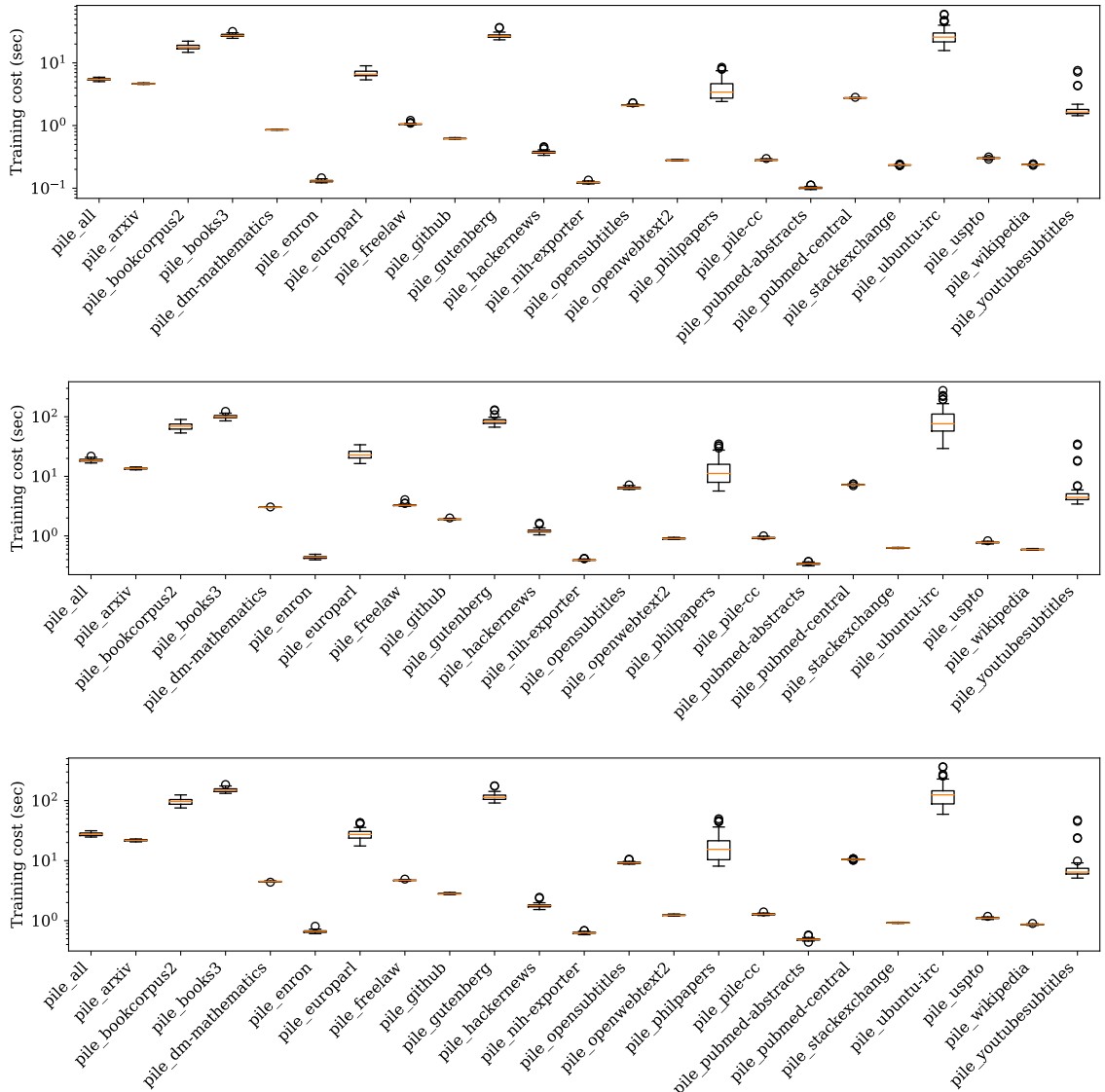

Figure 14: Training costs in seconds per neighbor. Top: GPT-2-Small. Middle: GPT-2. Bottom: GPT-Neo.

