# OpenReview forum: "Test-Time Training on Nearest Neighbors for Large Language Models"
_ICLR.cc/2024/Conference — ICLR 2024 poster_

### Official Review · Reviewer_y6wT · 2023-10-31

**Soundness:** 2 fair
**Presentation:** 3 good
**Contribution:** 2 fair
**Rating:** 5
**Confidence:** 5

**Summary:**

The paper proposes a retrieval-augmented LM recipe with these steps:

1) Given a query context, retrieve N sequences.
2) Sequentially finetune the LM on the N sequences.
3) Use the finetuned LM to predict the next tokens. (Although results in the main text use the same text for retrieval and testing)

The paper shows improve perplexity on multiple datasets of the pile, and uses the pile for retrieval (done with a distributed neighbor index). The main weaknesses of the paper are lack of analysis to show why retrieval is helpful (perhaps it only helps when finding exact matches in retrieval?) and weakly implemented retrieval-based baselines.

**Strengths:**

1. The approach is simple and effective.

2. There is evaluation using a large amount of data for retrieval, and across many datasets. Although, lack of analysis is concerning and makes it hard to understand how significant the improvements are.

3. The approach relies on a distributed neighbor index. In general, it will be helpful for the community working on retrieval-enhanced ML to see papers that operate with such large retrieval. Although there are not many details about how the distributed index is implemented besides the server count and amount of data used. "We find that 180 servers is a reasonable trade-off for the cluster we use." is not backed up by any statistics. Also, sometimes retrieval is done on this large scale with sparse retrieval, e.g. BM25, and dense retrieval is done here but not not with a model designed for retrieval such as DPR or others.

**Weaknesses:**

1. The paper is lacking in analysis. Consider this statement from the intro "In this paper, our hope is that a sufficiently large database will contain data relevant enough to each “domain”, induced by each test instance, such that fine-tuning improves local performance.". Perhaps it is worth measuring if the data found was from the matching domain? Although we see improvements in perplexity, we do not have a sense of why these improvements are happening. In addition, there is mention that 4 percent of retrieval is nearly an exact match---if the perplexity improvements is only due to exact match from the training data then it is not clear how useful this is.

2. The retrieval baselines are very weak and not well configured for this task. Also, "Unlike ours, models for those methods need to be trained also with retrieval." Except many of the models do not need to be trained with retrieval, including the ones used as baselines. Is the implication that models trained for retrieval are better than test time training?

2a. The kNN-LM comparison is far from fair. By looking at the interpolation param, the value is clearly much worse than expected. The param is only 0.02 but in the kNN-LM paper it is much higher 0.25 when the retrieved and test data are from the same domain, and 0.6 when they are from different domains. This is a huge difference and either the kNN-LM results should be excluded or amended. I make suggestions for how to amend the results in the questions section.

2b. The "in-context" baseline gives almost the same result as the base model. This suggests the in-context approach is almost not doing anything, probably for two reasons: 1) the context length is very limiting and 2) LMs often do not attend well to retrieved information in such large contexts (Sun et al and Liu et al). In contrast, Shi et al shows that when done properly the in-context approach can be very effective both for language modeling and QA. To be done properly, probably chunks should be retrieved and the query shortened. If there is still a context limit issue then Shi et al proposes an ensembling technique that enables scaling up the number of retrievals more efficiently.

3. Despite the clarification in 4.1 about retrieval-evaluation overlap, it seems inappropriate to include the results in Figure 5 as the model is using the text it is meant to predict for retrieval. Fortunately, this should be an easy fix as the fair comparison is already in the appendix.

4. In general, there are not many insights about how effective and influential retrieval is. It would be helpful to include an alternative retrieval, such as BM25. Similarly, it may be helpful to include an alternative dense retriever---since in this setup the LM is disjoint from the retriever, then it makes sense to include a model designed specifically for retrieval.

Sun et al. Do Long-Range Language Models Actually Use Long-Range Context?

Liu et al. Lost in the Middle: How Language Models Use Long Contexts

Shi et al. REPLUG: Retrieval-Augmented Black-Box Language Models

**Questions:**

Q1: What if the retrieved contexts are not relevant to the query context?

Q2: How would GPU acceleration further improve the speed? Isn't the data much too large for any GPU?

Q3: Is there any plan to release the code for the distributed server? Is it meant to be a novel contribution of this work? Is there existing work that achieves anything similar? My impression is that there are multiple options for this type of distributed neighbor index used in industry.

Q4: "largest gradient step" I am confused what is meant by this. I assume it is meant to correspond to learning rate scheduling, but in general I would think the furthest neighbors may have a larger gradient magnitude if they are more different from the existing context. Also, can you simply measure the gradients and see if this is true or not---whether the first step is the largest?

Suggestions for kNN-LM

* Use GPT2 for retrieval. This has been shown to work well with kNN-LM.
* Use only a single dataset, e.g. github, for retrieval and encode every token for all or contiguous subset of the data.
* Retrieve at every token, and follow the recipe from kNN-LM.
* Alternatively, simply report perplexity on a dataset that kNN-LM was already evaluated for.


Minor notes

* Fig 7: What are the top tasks? I assume they are the largest.

Other Related Work

* Basu et al: This paper does test-time training on other tasks and also presents a modern theoretical view on the value of test-time training.
* Drozdov et al: This paper improves upon the kNN-LM by adapting the interpolation to the quality of retrieval.
* Ram et al: An effective application of the in-context approach.

Basu et al. A Statistical Perspective on Retrieval-Based Models

Drozdov et al: When and how to rely on retrieval in the kNN-LM

Ram et al: In-Context Retrieval-Augmented Language Models

---

> ### Author Response · Authors · 2023-11-22
>
> We are really grateful for your very detailed feedback. Thank you! Next we address your concerns and questions.
>
> Concerns:
> 1. Our motivation is actually that each test instance is its own “domain”. This so-called “domain” (in quotes) is not defined as, for example, a specific task in the Pile like github, but the test instance itself. Therefore, Figure 3 and 4 already serve as analysis for this motivation. The improvements in perplexity, from our point of view, happens because there are sufficiently close training instances to this test instance. When we manually inspect the 4% of data that have a nearly exact match, the matches are not the same text as the queries. They are just different texts that seem to have the same style. Different passages from the same novel, for example, could easily exhibit this pattern. This is not a case of test set leakage.
> 2. The implication is that models trained with retrieved neighbors require a huge amount of extra resources for training. This is why we choose not to compare with them.
> - (a) Generally speaking, KNN-LM has substantial improvements for medium size datasets like Wikitext-103, but is extremely resource intensive, to the point of almost impractical, due to its token-level retrieval and database. Our method is practical for very large datasets. In fact, the larger the dataset, the more relevant our retrieved neighbors, and the better our method should be. Our baseline is meant to be a practical implementation on the Pile, which would otherwise be too large for KNN-LM. Following the recipe of KNN-LM would use hundreds of times more memory and compute. Downgrading to a smaller dataset like Wiktext-103 would force us into a regime where our method is not meant to be advantageous.
> - (b) The in-context baseline is an approximation to retrieval-augmented LMs if they are not trained with retrieval. The point is to put their training cost and ours into the same regime. On the other hand, we agree that the context length of Roberta and GPT-2 is limiting the performance of this baseline.
> 3. Thank you for the suggestion. We will highlight the results of the split evaluation.
> 4. Our choice of Roberta for retrieval (instead of GPT-2) is exactly motivated by the consideration that Roberta is more suitable for retrieval, since each output embedding takes advantage of the global context. We agree that other alternatives for building the index would be interesting. We chose to use the simplest possible option, to demonstrate the potential of this method if extended with more sophisticated techniques.
>
> Questions:
> 1. We have tried setting a distance threshold, above which we would not train on the retrieved text. In general, we find that a higher threshold makes performance worse, but could save a little time on average. We chose not to incorporate this technique in order to keep our method simple.
> 2. The FAISS library we are using comes with a GPU acceleration mode that has been tested on indices larger than ours. In their documentation, it seems to improve efficiency for retrieval.
> 3. All of our code, including that of the distributed server, and, in fact, the complete index have already been made publicly available. The URL was omitted to preserve anonymity during the reviewing process. It’s possible that people in industry already have their proprietary systems built. They are just not publicly available.
> 4. Earlier gradient steps in general have larger magnitudes than later steps, because the loss is high at the beginning of optimization. For our neighbors, we have observed that the initial loss is higher than the later ones, regardless of the ordering of the neighbors. This makes sense because the initial model was trained to be globally good on the entire dataset, so the local differences between the neighbors matter little for its initial loss.

---

> > ### Comment · Reviewer_y6wT · 2023-11-22
> >
> > Re: domain,
> >
> > > Our motivation is actually that each test instance is its own “domain”. This so-called “domain” (in quotes) is not defined as, for example, a specific task in the Pile like github, but the test instance itself. Therefore, Figure 3 and 4 already serve as analysis for this motivation.
> >
> > My initial concern holds. There is not enough analysis that explains why perplexity is improving. Figure 3 and 4 show statistics about neighbor distances, but is completely independent of perplexity improvement.
> >
> > > When we manually inspect the 4% of data that have a nearly exact match, the matches are not the same text as the queries. They are just different texts that seem to have the same style. Different passages from the same novel, for example, could easily exhibit this pattern.
> >
> > I do not understand how two texts can "nearly an exact match" and still have this property of being so different.
> >
> > Re: baselines,
> >
> > I remain very confused why kNN-LM was chosen as a baseline, given that it's clearly not well suited for this setting. In my initial review I suggested a setting where it would be more fair to compare against kNN-LM. I suggested an alternative baselines (like REPLUG) that is probably better than kNN-LM on your setting.
> >
> > Also,
> >
> > > One interpretation of this empirical phenomenon is that, it could be advantageous to take the largest gradient step at the beginning on the best available data point, and therefore ground the fine-tuning process in a better region of the loss landscape.
> >
> > As far as I can tell here, there is no guarantee about relative magnitudes of gradients, and it is all data dependent. I think intuitively, it may make sense to start with the closest neighbor because it may be more reliable than the furthest on average, but in general I am not convinced the first step will be largest gradient nor that the closest neighbor is the "best". I am nitpicking, but I think the reader may easily get confused by this and I suggest rewording to be more precise.
> >
> > Overall, I quite like this paper especially given the ambition wrt retrieval and amount of data, but I think the primary weaknesses mentioned in my initial review are the same after rebuttal: "The main weaknesses of the paper are lack of analysis to show why retrieval is helpful (perhaps it only helps when finding exact matches in retrieval?) and weakly implemented retrieval-based baselines." For now I plan to keep my score the same as weak reject.
> >
> > This makes it hard for me to trust the results even though it is a nice idea (similar to Basu et al, but with more detailed experiments on LMs).

---

### Official Review · Reviewer_zh5b · 2023-11-01

**Soundness:** 3 good
**Presentation:** 3 good
**Contribution:** 2 fair
**Rating:** 5
**Confidence:** 4

**Summary:**

This paper investigates test-time training on nearest neighbors (TTT-NN) in the context of large language models, specifically transformer models. The authors create a large-scale distributed nearest neighbor index based on text embeddings of the Pile dataset. For each test instance, the system retrieves nearest neighbors from this index and fine-tunes the model on these neighbors before applying it to the test instance. The method is evaluated on 22 language modeling tasks from the Pile benchmark, using three causal language models of increasing size (small GPT2, large GPT2, and GPTNeo).

The results show that training for only one gradient iteration on as few as 50 neighbors can reduce a normalized perplexity measure (bits per byte metric) by 20%. Test-time training narrows the performance gap between a small GPT2 model and a GPTNeo model, which was specifically trained to convergence on the Pile. The improvements due to TTT-NN are more dramatic on unseen tasks, while still helpful on seen tasks. Test-time training can increase the effective capacity of a model, though at the cost of increased inference time. The authors conclude that their work establishes a valuable baseline for implementing test-time training in large language models and opens the door to further research in this area.

**Strengths:**

1. The organization of this paper is well-structured, making it easy to read and comprehend.
2. This paper presents a simple test-time training approach on nearest neighbors (TTT-NN), which significantly improves performance across more than twenty language modeling tasks in the Pile benchmark with minimal fine-tuning.
3. Test-time training effectively increases the capacity of a model, showcasing its potential to narrow the performance gap between smaller and larger models, and offering a valuable  baseline for implementing test-time training in the context of large language models.
4. The large-scale distributed nearest neighbor index built on text embeddings of the Pile dataset enables efficient retrieval of relevant data for test-time training, serving queries to approximately 200 million vectors and 1TB of data in just one second.

**Weaknesses:**

1. Why not use the PQ (Product Quantization) Index, which can significantly reduce storage overhead and thus avoid the cost of distributed retrieval? Although the vectors after PQ are approximations of the original vectors, recent works such as ”**[KNN-MT](https://openreview.net/forum?id=7wCBOfJ8hJM)“** have demonstrated better performance using this approach.
2. Retrieval plus k*seq_len gradient updates may reduce inference speed. How much of a difference is there between the inference speed of the proposed method and the original model?
3. How is the database used by TTT-NN constructed? Is it built using the training data from Pile?
4. How does the baseline "interpolation with the distribution of tokens among the neighbors" work? What is the key used for retrieval when predicting the next token? Also, for KNN-LM, directly constructing the database is indeed very costly. Dai's work ”**[SK-MT/SK-LM](https://openreview.net/forum?id=uu1GBD9SlLe)“** provides an efficient construction method for their KNN-LM, i.e., first using BM25 to retrieve similar N documents, and then using these N documents to build a token-level database for interpolation and prediction. Considering that this work retrieves k documents, how are the k documents and the original model's predicted probability distribution interpolated? Is it similar to Dai's work mentioned above?
5. For Section 4.1, "Splitting sequences to avoid retrieval-evaluation overlap," suppose the test sequence is $x_t = ABCDEFGH$. Do the authors mean that there might be a sentence $x_d = ABCDEFGH$ in the database? If so, does this introduce test data leakage? Moreover, even if we split $x_t$ into $x_t^{'}=ABCD$, according to the description in Section 3, when using the prefix for retrieval, can we still retrieve $x_d$ and thus cause test data leakage?
6. What does "plain" refer to in Table 1? No specific definition was found.
7. Some missing related works
    - [REALM: Retrieval-Augmented Language Model Pre-Training](https://arxiv.org/abs/2002.08909)
    - [Training Language Models with Memory Augmentation.](https://aclanthology.org/2022.emnlp-main.382.pdf) The work is to retrieve similar k neighbors to aid training
    - To some extent, this paper can be considered as an explicit extrapolation of KNN-LM/KNN-MT. Specifically, Gao's work“[Nearest Neighbor Machine Translation is Meta-Optimizer on Output Projection Layer](https://arxiv.org/abs/2305.13034)” demonstrates that the working mechanism of KNN-LM/KNN-MT is to perform implicit gradient updates using the retrieved k nearest neighbors. In contrast, this paper explicitly uses the retrieved k nearest neighbors for explicit gradient updates.

**Questions:**

see above

---

> ### Author Response · Authors · 2023-11-22
>
> We really appreciate the detailed reviews. Thank you for your time! Here we respond to your concerns.
> 1. There are a lot of alternatives for building the index. We chose to use the simplest possible option, to demonstrate the potential of this method if extended with more sophisticated techniques.
> 2. Please see the table in Appendix E for details on inference speed. We're happy to highlight these numbers in the main body.
> 3. Yes, using the training data from the Pile.
> 4. We simply use the retrieved documents to get a distribution over tokens. The same distribution is used for all predictions of the next token in the same instance. The first paragraph on page 8 contains a detailed description.
> 5. The split evaluation is not for test data leakage in the training set. The purpose is to eliminate the possibility of using the target tokens as parts of the query for retrieval. Since the point of language modeling is to predict the next tokens as target, it could potentially be viewed as unprincipled to use those tokens for retrieval. The split evaluation avoids that.
> 6. In Table 1, plain means the baseline before TTT, i.e. direct inference with the pre-trained language model.
> 7. We have already cited Guu et al. (2020) for REALM. Thank you for the other two references, we’ll incorporate them.

---

### Official Review · Reviewer_Cq76 · 2023-11-01

**Soundness:** 4 excellent
**Presentation:** 4 excellent
**Contribution:** 3 good
**Rating:** 5
**Confidence:** 4

**Summary:**

This paper addresses the problem of improving the language model perplexities by using the training data during inference. The core idea is to find the sequences similar to the test sequence from the indexed training data and finetune the base model with these nearest neighbor sequences.

The paper evaluates this on pile benchmark, where the training data is indexed using representations obtained from the Roberta model. For each test sequence from various pile categories, the nearest neighbors are picked to finetune the model and later the test sequence is evaluated for perplexity measure. The empirical results show usefulness of the approach as it improves the LM perplexities.

**Strengths:**

1. The paper is well-written and easy to follow. The idea is clearly described and empirically validated.

2. The empirical results on various pile benchmark show the usefulness of the approach.

**Weaknesses:**

1. While the idea is neat and simple to implement, as shown in Figure 9, the training costs for each neighbor is expensive, thus limiting the usefulness in real-time applications.

2. While the results on the LM perplexity are useful, it would be interesting to see how this compares in an end-to-end task such as code generation, etc. Few-shot prompt tuning (with or without retrieval augmented learning) are popular paradigms that are used in bigger LLMs. It would be interesting to see the comparison with such methods (in offline evaluation settings).

**Questions:**

1. It is not clear to me how indexes handle larger sequences? Bigger sequences are chunked [chunk1, chunk2, chunk3, ..] and if the nearest neighbor match happens at chunk2, what is the process?

2. While it is neat that this method doesn't require hyper-paramter tuning? What happens when one tries that? (I agree it is prohibitively expensive, but could be done for few test sequences)

3. How does KNN-LM work with document level index? For the original work, it was context -> next_word, how do we get token probabilities with document level index.

**Details Of Ethics Concerns:**

Doesn't require ethics review.

---

> ### Author Response · Authors · 2023-11-22
>
> Thank you for your thoughtful reviews. Here we respond to the specific concerns and questions.
>
> Weaknesses:
> 1. While our method is more expensive than direct inference without retrieval, it is less expensive than retrieval-augmented LMs (see in Section 2.2), which is one of the baselines.
> 2. We agree that few-shot evaluations would be nice.
>
> Questions:
> 1. For long sequences, we only use chunk1 for retrieval, as explained in the first paragraph on page 4. Specifically, chunk1 contains the maximum number of tokens that can fit into the context window. Using a more elaborate scheme would probably improve performance even further.
> 2. If you find our current results convincing, then the lack of hyper-parameter tuning means that they could be even better. So the method might have still more potential than it appears in the results.
> 3. We simply use the retrieved documents to get a distribution over tokens. The same distribution is used for all predictions of the next token in the same instance.

---

### Official Review · Reviewer_kVHL · 2023-11-02

**Soundness:** 3 good
**Presentation:** 2 fair
**Contribution:** 3 good
**Rating:** 6
**Confidence:** 4

**Summary:**

This paper proposes a method of training at test time

**Strengths:**

* The method is not too complicated, and could likely be reproduced.
* In some ways, the evaluation was very impressive. Quite large scale, showing benefits with an index that spans the whole Pile across many domains.
* The baselines of kNN and in-context prompting also seemed relevant/strong.

**Weaknesses:**

There were some weaknesses. I think this paper still could have value, but I would be more confident in recommending that the paper be accepted if the following could be addressed:

1. There are some clarity issues with the paper. For instance, it was not very clear to me if retrieval is done after every token or at some other cadence.

2. There is a discussion of inference speed, but it is not very concrete. Could inference throughput be added to table 1?

3. While bits/byte based LLM evaluation is good, it would also be really nice to see results on extrinsic tasks as well.

4. This is not so much a weakness as a missed reference, but this paper is very relevant: https://arxiv.org/abs/1609.06490
Li, Xiaoqing, Jiajun Zhang, and Chengqing Zong. "One sentence one model for neural machine translation." arXiv preprint arXiv:1609.06490 (2016).
I think it should definitely be cited, but I think even if a similar idea has been proposed before in the neural (conditional) LM space, the modernized evaluation of the current paper has significant value.

**Questions:**

See weaknesses above.

* Also, will the code/data/datastore be released so others can reproduce these studies?

---

> ### Author Response · Authors · 2023-11-18
> **Thank you for your helpful feedback**
>
> We're grateful for your helpful feedback. Here are responses to the specific items:
>
> 1. Retrieval is done once based on a prefix of the query. There is no retrieval after each token.
> 2. Please see the table in Appendix E for details inference speed. We're happy to highlight these numbers in the main body.
> 3. We agree that additional evaluation is a good step for future work.
> 4. Thanks for the pointer. We'll incorporate the reference.
>
> All of our code and, in fact, the complete index have already been made publicly available. The URL was omitted to preserve anonymity during the reviewing process.

---

### Meta-Review · Area_Chair_PbRc · 2023-12-09

**Metareview:**

This paper proposes a method to finetune a model on retrieved examples (using a nearest neighbors index with test embeddings) at test time. Detailed experiments are presented on the large Pile collection, showing that the method while more expensive than typical inference (e.g., few-shot prompting) is more efficient than other retrieval-augmented language modeling approach.

**Strengths**: The greatest strength of the paper is the simplicity of the idea, combined with very strong empiricism. Large quantities of data were used for building the index and results are presented across multiple settings. This work could be impactful in safety critical domains, such as clinical knowledge / medicine.

**Weaknesses:** There remain some outstanding questions on the speed of inference and the practicality of the approach. How reusable are test-time models? The efficacy of this somewhat expensive approach to increase model capacity needs to be studied on downstream tasks? Some relevant related work pointed out by the reviewers seem to have been missed. Somewhat important details about splitting / chunking the query and the index were not clear.

**Justification For Why Not Higher Score:**

See weaknesses above, some missing details and practicality concerns gave the reviewers some pause.

**Justification For Why Not Lower Score:**

I believe this work deserves being accepted to ICLR because the method proposed is somewhat surprisingly simple while underexplored. The empiricism of the work makes it clear that much effort has been invested into the idea.

---

### Decision · Program_Chairs · 2024-01-16

Accept (poster)